# Unexpected Classes of Aquaporin Channels Detected by Transcriptomic Analysis in Human Brain Are Associated with Both Patient Age and Alzheimer’s Disease Status

**DOI:** 10.3390/biomedicines11030770

**Published:** 2023-03-03

**Authors:** Zein Amro, Matthew Ryan, Lyndsey E. Collins-Praino, Andrea J. Yool

**Affiliations:** 1School of Biomedicine, University of Adelaide, Adelaide, SA 5005, Australia; 2School of Mathematical Sciences, University of Adelaide, Adelaide, SA 5005, Australia

**Keywords:** water channels, peroxiporins, ageing brain, dementia, transcriptomics

## Abstract

The altered expression of known brain Aquaporin (AQP) channels 1, 4 and 9 has been correlated with neuropathological AD progression, but possible roles of other AQP classes in neurological disease remain understudied. The levels of transcripts of all thirteen human AQP subtypes were compared in healthy and Alzheimer’s disease (AD) brains by statistical analyses of microarray RNAseq expression data from the Allen Brain Atlas database. Previously unreported, AQPs 0, 6 and 10, are present in human brains at the transcript level. Three AD-affected brain regions, hippocampus (HIP), parietal cortex (PCx) and temporal cortex (TCx), were assessed in three subgroups: young controls (*n* = 6, aged 24–57); aged controls (*n* = 26, aged 78–99); and an AD cohort (*n* = 12, aged 79–99). A significant positive correlation (*p* < 10^−10^) was seen for AQP transcript levels as a function of the subject’s age in years. Differential expressions correlated with brain region, age, and AD diagnosis, particularly between the HIP and cortical regions. Interestingly, three classes of AQPs (0, 6 and 8) upregulated in AD compared to young controls are permeable to H_2_O_2_. Of these, AQPs 0 and 8 were increased in TCx and AQP6 in HIP, suggesting a role of AQPs in AD-related oxidative stress. The outcomes here are the first to demonstrate that the expression profile of AQP channels in the human brain is more diverse than previously thought, and transcript levels are influenced by both age and AD status. Associations between reactive oxygen stress and neurodegenerative disease risk highlight AQPs 0, 6, 8 and 10 as potential therapeutic targets.

## 1. Introduction

Healthy aging is associated with widespread cognitive, morphological, and functional changes in the brain. Such processes are exacerbated in age-related neurodegenerative disorders, including Alzheimer’s disease (AD) [1]. AD, characterized by the formation of amyloid plaques and neurofibrillary tangles (NFTs) consisting of hyperphosphorylated tau in vulnerable brain regions, is the leading cause of dementia in the aging population [2]. Amyloid plaques are thought to impair synaptic function, induce hyperexcitability, and enhance the generation of reactive oxygen species [3,4,5,6]. Similarly, insoluble NFTs of hyperphosphorylated tau have been correlated with neuronal toxicity [7], and found to serve as predictive markers for cognitive performance and overall dementia status [8]. Importantly, amyloid beta (Aβ) and tau-based evaluations indicate that the disease spreads through neighbouring anatomical areas beginning at the hippocampal formation and areas of the temporal (e.g., entorhinal cortex) and parietal (e.g., retrosplenial cortex; posterior parietal cortex; precuneus) lobes in preclinical stages of the disease before spreading to additional regions (e.g., prefrontal cortex; amygdala) as individuals become symptomatic [9,10,11] in a process proposed to involve networks of astrocytes and microglia [12]. In particular, the astrocytic internalization of Aβ plaques for clearance in AD has been suggested to involve aquaporin channels [13,14,15].

Aquaporin channels (AQPs) are transmembrane proteins that facilitate the bidirectional movement of water and small solutes, and are expressed in all forms of life [16,17]. The 13 classes of AQPs in humans (AQPs 0–12) show tissue-specific expressions in brain, kidney, eye, skin, heart, lungs and other organs [18]. The classical AQP subtypes, initially defined as strictly water-selective, include AQPs 0, 1, 2, 4 and 5, though additional permeabilities to ions, signalling molecules and metabolites continue to be added to the repertoire [19,20]. For example, AQP0 and 5 have both been shown to be permeable to hydrogen peroxide (H_2_O_2_) in addition to water [21,22], prompting an additional descriptor as ‘peroxiporins’. Similarly, AQPs 6, 8 and 11 are all classified as peroxiporins [23,24,25]. AQP subtypes initially characterized by their permeability to both glycerol and water are classified as aquaglyceroporins, including AQPs 3, 7, 9 and 10 [26], although AQP9 has also been shown to permeate H_2_O_2_ in mice [27]. Finally, the non-orthodox AQP12, which, similar to the peroxiporin AQP11, lacks one of the two conserved asparagine-proline-alanine (NPA) motifs important for the molecular structure of the pore passageway [28], has been suggested to play a role in digestive enzyme secretion [29]. 

In the mammalian brain, three AQPs—AQP1, 4 and 9—have been identified as proteins expressed under physiological and pathological conditions [30,31,32,33]. AQP1 is predominantly expressed in the choroid plexus, facilitating regulated cerebrospinal fluid (CSF) production under normal physiological conditions [30]. In the face of pathology (such as AD, contusion and subarachnoid haemorrhage, for example), reactive astrocytes initiate the abnormal expression of AQP1 [31,34]. In line with this, AQP1 levels are increased in early AD (as defined by Braak criteria) within astrocytes, and co-localized with Aβ plaques for reasons yet to be determined [35,36,37]. Unlike AQP1, which only is expressed in astrocytes under pathological conditions, AQP4 is dubbed the ‘brain AQP’ based on its high levels of physiological expression in astrocytes throughout the central nervous system, primarily in perivascular and peri-synaptic end-feet domains [32,38]. AQP4 channels are essential for the clearance of interstitial solutes, metabolic products and protein aggregates (such as Aβ and hyperphosphorylated tau) from the brain microenvironment via the glymphatic system [15,39,40,41,42,43]. Similarly, AQP9 is also thought to play a key role in astrocytes under normal physiological conditions and has been suggested to be involved in facilitating the diffusion of lactate from astrocytes to neurons for metabolic support [33,44,45].

Other AQPs also have been implicated in brain function under both normal physiological and pathological conditions, although, to date, these have been less well investigated. For example, AQP6 has been suggested to participate in the gated reabsorption of water to reduce neuronal synaptic swelling, since its expression and activity are reliant on low pH [46]. Transcript and protein levels of AQPs 3, 5, 8 have all been reported to increase in rat astrocytes and neurons in vivo after hypoxia, suggesting that they may play a possible role in post-injury edema (40). Interestingly, AQP11 has been detected in cerebellum and hippocampus, but no functional role of this non-orthodox AQP has yet been proposed [47,48]. 

Given the diverse expression of AQPs within the mammalian brain, and their multi-faceted roles in numerous physiological processes, we propose that alterations in AQP channel expression may play a role in both healthy aging and AD pathogenesis. Significantly, given their role in response to stress and injury, we hypothesized that establishing AQP expression profiles in AD might reveal their potential as novel therapeutic targets. To identify the full set of candidate AQPs of interest, transcript levels were assessed from Allen Brain Atlas data for all 13 AQP subtypes, in brain regions selected for relevance in AD pathology. Transcript levels in AD brains were compared with age-matched healthy brains and young healthy controls. Notably, the Allen Brain Atlas serves as a substantial archive of collated RNAseq data that remain to be analysed; this public domain database is invaluable for enabling novel discoveries, as demonstrated in previous work [49].

## 2. Materials and Methods

### 2.1. Data Source 

Data were acquired from the Allen Brain Atlas (ABA), a publicly available dataset. Microarray data were downloaded directly from the webpage (https://human.brain-map.org accessed on 21 September 2021). In brief, brains were processed serially with multiple sample batches submitted per brain analysed. Data were then normalised to an internal control, in accord with detailed method documents (https://help.brain-map.org/display/humanbrain/Documentation accessed on 21 September 2021).

### 2.2. Human Brain Atlas Database 

This dataset contains RNAseq transcriptome data from six individuals aged 24–57 with no known pathology, designated as the young control group (C). Data from this database, in addition to the Institute of Aging, Dementia and TBI database, were collected from four brain regions for all age groups (Appendix A), given the anatomical areas of interest affected in AD highlighted in the introduction and as outlined by the Braak staging for disease spread [9]. Given the cortical areas known to be affected early on in the disease process in AD, the current analysis focused specifically on data from the temporal and parietal lobes [11]. A detailed description of tissue acquisition is available in the ABA white paper documentation (https://help.brain-map.org/display/humanbrain/Documentation accessed on 21 September 2021). Briefly, brain tissue was collected after obtaining informed consent from the patients’ next-of-kin, followed by a review and approval from the Institutional Review Board (IEB).

### 2.3. Institute Aging, Dementia and TBI Database 

This dataset contains RNAseq transcriptome data from 107 individuals aged 77 and older with/without traumatic brain injury (TBI) and dementia obtained from the Adult Change in Thought cohort [50]. To investigate the effect of aging on the AQP gene expression profile in the brain, an aged control group (AC), comprised of 29 individuals aged 78–99 with no known pathology, was used for comparison with both the C and AD groups. For the AD group, 12 individuals aged 79–99 with a pathological diagnosis of probable AD and no prior history of TBI were selected for analysis. Individuals with a diagnosis of possible AD were excluded, as their underlying disease progression may be secondary to other comorbidities [51]. Additionally, due to the known influence of TBI on tau pathology and its relationship with an increased risk of AD [52,53], patients with any documented history of TBI were excluded from bioinformatics analysis. 

### 2.4. Gene Probes 

For each AQP channel gene, two probes for the young control group (selective for different exons) and one probe for the aged control and AD group were used for RNAseq analysis. For details on probe IDs, refer to Appendix A. For a comparison between groups, the two probes used for each gene in the C group were averaged.

### 2.5. Statistical Analysis 

#### 2.5.1. Regression Model Analyses 

To investigate the relation between age and AQP0-12 RNAseq levels, we used a random intercept model generated using the formula:$${\mathrm{RNAseq}\;\mathrm{level}}_{gene,region}=\beta_{0,gene,region}+\beta_{1}age+\varepsilon \;.$$

Additionally, linear regression models were fit for all subjects for each gene/anatomical region separately, controlling *p*-values using the Bonferonni correction.

#### 2.5.2. Supervised Clustering Analyses 

Using our genes and anatomical regions of interest, supervised clustering methods were used to investigate which of the probes were primarily responsible for differences between clusters determined in the healthy young and aged groups (https://human.brain-map.org accessed on 21 September 2021). This method produces Principal Components, defined as a function of the probes loaded into the analyses. The ‘loadings’ (coefficients) for the probes show how strongly each of the probes affect the clustering. Each principal component describes a linear combination of probes that best distinguishes between the three anatomical regions (HIP, PCx, TCx). Probes for AQP11 and AQP9 defined Principle Component 1. A total of 22 probes for various AQP genes defined Principle Component 2. Supervised clustering analysis was performed using Sparse Partial Least Squares Discriminant Analysis, as implemented in the *mixOmics* package [54] on RStudio. 

#### 2.5.3. Differential Expression Analysis 

To determine whether there was a differential expression between each of the three anatomical groups of interest in the young control (C) and AC group pooled, a differential expression analysis was conducted using the *limma* package [55] to fit linear models on RStudio. The models included individual ID as a covariate in order to account for the nested structure of the data, which included multiple samples from each of the six individuals. Furthermore, to investigate whether AQP gene expression changes with healthy aging, a differential expression analysis was conducted on these individuals, dividing them into their original groups (C and AC). Heatmaps were generated by graphing the log fold change (logFC) of genes on GraphPad Prism 9.0 for probes that both showed a significant result in the differential expression analysis.

#### 2.5.4. Expression Analysis–Group Comparison 

To investigate the potential change in individual gene expression profile in the control, aged control and AD groups, a one-way ANOVA followed by multiple comparison post hoc Tukey test of the RNAseq expression level for each AQP channel was conducted on GraphPad Prism V9.0. The significance level for all analyses was set at *p* < 0.05.

## 3. Results 

### 3.1. Subject Population Characteristics

The demographic information for each group is presented in Table 1. No statistically significant difference between the Aged Control (AC) and Alzheimer’s disease (AD) groups was observed for education level (*p* = 0.655) but was observed for age when comparing AC and AD groups to the C group (*p* < 0.0001). However, as expected, AD patients showed a significantly advanced degree of pathology, as measured by the Braak stage (*p* < 0.05). Detailed patient information is summarized in Appendix A. 

### 3.2. Baseline AQP Expression Profiles Differ with Age in the Healthy Brain

Patterns of AQP4 channel expression in the brain have previously been shown to change during healthy aging [56]. Work here investigated the association of age with the expression profile of all AQPs in selected brain areas known to be impacted by AD, namely, the hippocampus (HIP), parietal cortex (PCx) and temporal cortex (TCx). In the C and AC groups, a supervised cluster analysis was conducted to probe the relationships between age and AQP baseline expression profiles (Figure 1). Using a mixed-effect linear plot to test for an overall relationship, a significant positive correlation (*p* < 10^−10^) was observed for AQP RNAseq transcript levels as a function of the subject’s age in years (Figure 1a). When segregated by AQP channel subtype, interesting region-specific differences in the age-dependence of expression were evident in the linear regression plots (Figure 1b). AQP1 and 4, previously identified in the human brain, show a significant upward trend of expression in the HIP only as a function of age (*p* < 0.0001). AQP5 and 10 gene expression profiles also increased with age in the HIP and PCx, respectively (*p* < 0.05), the novel AQP channels not previously identified in the human brain. AQP9 shows an upward trend in expression within the HIP with age but with no significance, rather, a significant downward expression is evident in both cortical regions investigated, PCx and TCx, with age (*p* < 0.05). 

### 3.3. AQP Expression Profiles in the Hippocampus Differ from Those in Cortex in Healthy Brain

A clear distinction in expression profiles was observed between the HIP cluster and the PCx and TCx clusters, with no difference between cortical regions observed when comparing probes of AQP channel genes within each anatomical region (Figure 2). Two principal components (Figure 2a, component 1 and component 2) were evaluated as a function of all probes used (Appendix A). The distinction between the observed clusters was defined almost entirely by probes 1059114 (AQP9) and 1032651 (AQP11), suggesting an important role of these two AQPs in driving the differences between anatomical regions (Appendix A). Subsequently, a differential expression analysis was used to determine the log fold change (logFC) of gene expression within the PCx and TCx regions as compared to the HIP region (Figure 2b). Interestingly, for all AQP probes tested, the PCx and TCx regions showed no difference in expression profiles when compared to each other (Figure 2b,c); in contrast, differences were observed when either cortical region was compared to the HIP (Figure 2b,c). AQP11 probes showed significantly higher logFC values (*p* < 0.001; Table 2) in both PCx and TCx when compared to HIP (Figure 2d). Conversely, probes for AQP1, 3, 4, 9 and 10 showed low logFC values in the cortical regions, and higher expression in the HIP (Figure 2d, Table 2). Results here suggest that the diversity of AQP channels present in the human brain is broader than previously reported, and that their levels differ based on the anatomical location, as illustrated here for the HIP and cortical regions. 

### 3.4. Age-Dependent Changes in AQP Expression Profiles Differ between the HIP and Cortical Regions in the Healthy Brain

Differences in expression profiles of AQPs detected in the HIP as compared to the PCx and TCx (Figure 2) were investigated for effects of the subject’s age in healthy brains from all ages (C and AC) (Figure 3). A comparison of transcript levels looking at the direction of gene expression change (Figure 3a) between each cortical region and the HIP showed that the logFC values of AQPs 1, 4 and 5 were lower in cortex in AC but not in C groups (Figure 3b,c). Conversely, logFC for AQP10 was lower in both cortical regions as compared to the HIP in the C but not the AC group. Interestingly, the expression profile of AQP9 in the PCx decreased with age, shifting from levels higher than the HIP in the C group to lower than HIP in the AC group (Figure 3b). In the TCx, a similar shift in the pattern of expression of AQP9 was observed, ranging from no difference as compared to the HIP in C to lower levels of expression in AC (Figure 3c). Regardless of age, both cortical regions showed lower levels of AQP3 (*p* < 0.001) and higher levels of AQP11 (*p* < 0.001) compared to the HIP (Figure 3b,c). 

Analyses of the differential expression for AQPs in the parietal cortex versus hippocampus (Table 3) and in the temporal cortex versus hippocampus (Table 4) for both the C (i) and AC (ii) groups showed that most AQP classes (AQPs 0, 2, 6, 7, 8, 12) maintained comparable expression levels across both cortical regions as compared with hippocampus in healthy controls, and did not appear to be affected by age. Regional differences that appeared insensitive to age were observed for AQPs 11 and 3; AQP11 was consistently higher in both PCx and TCx than HIP in both young and aged cohorts, and *AQP3* was consistently lower in the cortex than hippocampus across both age groups (Table 3 and Table 4). Regional differences that were sensitive to age were observed for AQPs 1, 4, 5, 9 and 10. In this set, lower levels of transcripts in both cortical regions were observed in the aged but not the young cohorts for AQPs 1, 4, 5, 9. Conversely, AQP10 was lower in the cortical regions than the hippocampus in young cohorts, but there was no difference between regions in the aged cohorts. These data suggest that aging has a notable effect on the expression profiles of several AQPs, but not AQPs 3 and 11 (as highlighted in Figure 3 above). Differences in levels of transcripts between anatomical regions suggest specialized roles or distributions for AQP classes among neuronal and glial cell types, supporting the idea that the levels and patterns of AQP expression also might be sensitive to age-related disease states, such as AD.

### 3.5. Regional Differences in Levels of AQP Transcripts Associated with Probable Alzheimer’s Disease

In the probable Alzheimer’s disease cohort, region-specific subsets of the classes of AQPs showed higher levels that were significantly greater or showed trends towards elevations in the disease group that exceeded the levels observed in aged controls (Figure 4). The AD-associated trends toward augmented levels of transcripts were observed in the temporal cortex for AQP0, in the parietal cortex for AQPs 5 and 10, and in the hippocampus for AQPs 1, 4, 5, 6 and 9 (Figure 4a). The reverse trend in which transcript levels decreased in AD as compared with AC was seen uniquely for AQP7 in the hippocampus.

Another intriguing pattern that emerged from this analysis was for AQP9, with levels high in young controls, substantially reduced in both AC and AD cohorts for both cortical regions, and conversely elevated with age in the hippocampus (Figure 4a). There were no changes in AQP3 or AQP11 levels in AD as compared to AC and C groups within regions, although differences between regions were observed. AQP3 was predominantly in the HIP with little cortical expression. Conversely, AQP11 levels were high in both cortical regions but minimal in the HIP. Consistent baseline levels of expression of AQPs 3 and 11 suggest that aquaglyceroporin function in the hippocampus and peroxiporin activity in the cortex are ongoing mechanisms of metabolism and homeostasis. With the notable exception of reduced cortical AQP9, the expression profiles for AQP channels increased during natural aging, a process that in a subset of AQP classes appeared to be amplified by AD pathology.

A limitation of this study was the low n values available for human samples; the transcript levels for AQPs 0, 6, 8 and 10 showed increases that only reached statistical significance in the AD group when compared to the C group (Figure 4a). For example, in HIP, the AQP6 transcript was increased in the AD as compared to the C group (*p* < 0.05); AQPs 0 and 8 expression levels were higher in the TCx of AD patients than C patients (*p* < 0.05); and AQP10 expression was increased in AD patients in the PCx (*p* < 0.001; Figure 4a) as compared to the young control group. However, it is important to note that when directly comparing AD to age-matched AC groups, there were no significant differences in transcript levels for any classes of AQPs.

Nonetheless, potentially informative patterns of increased AQP levels in AD were apparent for specific classes, as observed from data compiled as plots of average logFC values for the AD cohort versus the AC group (Figure 4b) in each of the three brain regions. In this analysis, identical levels of expression produce a theoretical line with a slope of 1.0. AQPs with levels higher in AD than AC are reflected by points above the line. For example, in the HIP, the average trend points of AQPs 5, 6, 9 and 11 fell above the line. In the PCx, AQPs 0, 3, 7, 9 and 11 were higher in the AD cohort. In the TCx, AQPs 0, 1, 7 and 11 were higher in the AD versus the AC cohorts (Figure 4b). While not definitive, these results offer testable predictions for ongoing work aimed at gaging the potential importance of the variety of CNS-expressed AQP channels (such as AQPs 0, 7 and 11 in cortex) as targets of interest for understanding healthy functioning, as well as processes of neuropathology.

## 4. Discussion

We have discovered a surprising diversity of AQP channels in the CNS, confirming seven known classes of AQPs, and showing that three additional classes—AQPs 0, 6, and 10—are present in the human brain at the transcript level. Prior work defined three classes of AQP channels, establishing AQPs 1, 4 and 9 as the primary subtypes expressed in the mammalian brain in both physiological and pathological conditions [30,33,42]. Less well investigated RNA signals for other AQPs such as AQPs 3, 5, 8 and 11 also have been detected in the brain in vivo [47,48], though possible functions remain to be defined. Work here using data harvested from the Allen Brain Atlas explored the expression profiles of all classes of AQPs (AQPs 0–12) in the human brain, and investigated whether their expression patterns were affected in healthy aging and AD. We identified novel RNAseq signals for AQPs 0, 3, 5, 6, 7, 8, 10 and 11 in the human brain in hippocampal and cortical regions known to be impacted by Alzheimer’s disease. Of these channels, peroxiporins AQPs 0, 6 and 8 [21,23,24] (which are permeable to H2O2) and the aquaglyceroporin AQP10 (permeable to glycerol) were expressed at higher levels in AD as compared to young controls. 

The major advance reported in this study is the demonstration that AQP channels shown previously to be permeable to H_2_O_2_ (termed ‘peroxiporins’) show subtype-specific patterns of expression in the human brain that vary as a function of age, neuroanatomical region, and Alzheimer’s disease status. Oxidative stress levels are known to increase during aging [57], resulting in increases in reactive oxygen species by-products, such as H_2_O_2_. H_2_O_2_ levels are further elevated in AD as compared to healthy aging brains and are thought to potentiate mitochondrial dysfunction and disease pathology by promoting Aβ-induced neurotoxicity and pathological tau modifications [58,59,60]. The second outcome of interest here is the finding that multiple classes of aquaglyceroporins are differentially regulated with respect to brain age and disease status.

In the hippocampus in particular, a subset of the AQP classes (AQPs 1, 4, 5 and 9) showed strong increases with age, with or without AD. Additionally, in the hippocampus, AQP6 was increased in AD, and AQP7 showed higher levels with age. AQPs 7, 9 and 10 are aquaglyceroporins (discussed in more detail below). AQP1 has been shown to function as a perioxiporin in cardiac ventricular muscle cells [61]. Healthy aging previously was reported to correlate with the increased expression and localization of AQP4 in astrocytes [62]. AQP4, as do most members of the broad family, functions as a water channel but was not found to mediate H_2_O_2_ permeability when tested in the yeast expression system [63]; however, it is worth noting that AQP1 tested in the same assay similarly but did not enable detectable H_2_O_2_ fluxes, though this functionality was subsequently confirmed in mammalian heart cells [61]. AQP5 shows peroxiporin activity in the eye (21). Three classes of AQPs that showed increased levels in brain regions only in the presence of Alzheimer’s disease (AQPs 0, 6, 8) also are known to function as peroxiporins [21,23,24]. AQP0 has been characterized as an intrinsic membrane protein uniquely expressed in the eye lens and has been shown to facilitate transmembrane fluxes of H_2_O_2_ [21,64]. AQP0 expression in the brain is a novel finding. In the hippocampus, AQP6, which has been characterised as a peroxiporin in malignant pleural mesothelioma [23], also was higher in the AD cohort. AQP8, a pancreatic β-cell peroxiporin, similarly was detected at high levels in the TCx of AD patients. AQP11 showed a unique pattern in being expressed at higher levels in the cortex than in the hippocampus. AQP11 has been characterized as a peroxiporin in endoplasmic reticulum that mitigates H_2_O_2_-induced stress in the kidney proximal tubule cells [25]. The demonstration here of the AQP11 expression in the cortex and hippocampus, coupled with prior work confirming the AQP11 RNA expression in the cerebellum of mice [48], suggests that AQP11 might also be involved throughout the brain as one of the mechanisms involved in decreasing oxidative stress. 

Aquaglyceroporins that increased with age included AQPs 7 and 9 in the hippocampus, prompting the idea that an increased expression could be an adaptive response to altered metabolic demands [65]. In the parietal and temporal cortices, the changes in *AQP* expression with age were more subtle, ranging from no appreciable change to increased levels with age, with the exception of *AQP9* which conversely showed strong decreases in the Ptx and Ctx regions of aged brains. Neuronal ATP production is thought to decline with aging [66], leading to a hypometabolic state [67] which might be offset in part by enhancing the glycerol uptake to boost pyruvate production and ATP synthesis [68]. AQP9 in astrocytes is known to facilitate glycerol shuttling from astrocytes to neurons for energy support [45,69]. Data here suggest that *AQP9*, while increased in HIP, might be less likely to be a candidate for compensatory mechanisms in the cortex given the striking decrease in transcript levels observed in the cortical regions for both the AC and AD cohorts. In contrast, the aquaglyceroporin *AQP10* [70] was increased in the PCx of AD patients. AQP10 has been demonstrated previously to mediate the pH-sensitive transport of glycerol in adipose cells and enterocytes [70], and could serve a comparable role in the brain. Differences in the gating mechanisms between aquaglyceroporin classes might influence which subtypes are selectively upregulated to meet different brain region demands. The disease-specific increases in expression in certain AQP subtypes, which are statistically significant as compared to young controls, support a proposed association with neuropathological disease. 

An intriguing concept which we propose merits further study is that specific classes of peroxiporins might be upregulated as a protective mechanism to shuttle excess H_2_O_2_ and alleviate stress. The regional influences governing AQP expression patterns remain to be determined, but could reflect heterogeneity in neuronal and glial subtypes, differences in neuronal activity and metabolic demands or other factors. A single-cell RNAseq study in C57BL/6J mice by Batiuk and colleagues (2020) showed that astrocyte populations from the hippocampus of mice, unlike cortical regions, contained large numbers of progenitor astrocytic stem cells (AST4) as well as mature astrocytes (AST1) [71]. The AQP expression in specific astrocyte subpopulations such as those in the hippocampus [32,38] could explain in part the observed regional specialization of AQP expression patterns. Reactive astrocytes accumulate in regions of damage, including those affected in age-related cognitive decline, resulting in hypertrophy and cellular volume increases [72] which might be linked to AQP expression [73]. AQP1 and 4 expression levels are increased in astrocytes during the early pathology stage of AD [36,37]. With the continued expansion of the Allen Brain Atlas database, future work will benefit from comparing the transcript levels of AQPs between Alzheimer’s disease and age-matched controls. 

The limitations of this work are the modest *n* values available for human brain RNAseq data, which likely contributed to the lack of significant differences in expression profiles between the AD and the AC cohorts (though trends towards increased RNAseq levels were apparent in the disease group), and that age-dependent effects on the AQP expression are likely to overlap with the disease pathology. Another important limitation that might have influenced the absence of statistically significant differences between the AD and AC groups was the low representation of female donors in the AD group (*n* = 3) as compared to AC (*n* = 11). The risk for the development and progression of AD in females on average is higher than males but depends on estrogen levels [74]; a re-analysis of the AD and AC groups segregated by gender and hormone therapy status might reveal important correlations with AQP expression profiles that merit exploration when expanded database information becomes available. An additional limitation is the need to confirm AQP expression at the protein level, to determine whether the expression profile changes determined by transcript analyses are reflected by changes at the protein level. Probing the functionality of the proteins then will be an essential next step towards identifying possible novel targets for therapeutic interventions in AD.

It will be of interest to determine whether the trends towards similar responses observed here for AC and AD (which did not reach statistical significance) reflect processes of natural aging effects on AQP expression levels that are similar or amplified in the disease state. Histological analyses of human AD sections from the hippocampus have shown that the AQP1 expression appears to be localized in multipolar fibrillary astrocytes surrounding neurons, whereas AQP4 expression appears to be more diffusely distributed in astrocytes (34), suggesting that some of the spatiotemporal changes in *AQP* expression noted here might reflect changes in the regional status of astrocyte populations. It is important to consider that AQP up- or downregulation responses might be relevant to nervous system protection rather than being involved in driving the pathology, and levels of transcripts do not necessarily correspond directly to levels of functional protein in cell membranes. However, our data suggest that changes in AQP levels could be a response to natural processes of aging and mechanisms of either protection or pathology in neurodegenerative disease.

The patterns of AQP regulation in AD are novel and subtype-specific. The three established classes of brain AQPs described previously in normal physiological conditions, AQPs 1, 4 and 9, are associated with age, but data here suggest that these subtypes alone might not be sufficient to mediate responses to augmented pathological stressors. Our findings suggest that a diverse array of peroxiporin and aquaglyceroporin subtypes could be relevant to the processes of brain aging and disease. The results here are the first to show that AQPs 0, 6, 8 and 10 are expressed in the brain and increased with AD or age. Corresponding changes in protein levels and patterns of localization in neurons and glia remain to be defined and are a focus of work in progress. The exciting discovery of previously undetected classes of peroxiporins and additional aquaglyceroporins in the human brain compels further research on their potential roles in aging and AD-related diseases. Understanding the roles of an expanding array of identified brain peroxiporins and aquaglyceroporins in the brain is needed for uncovering homeostatic mechanisms that enable healthy aging and protection from damage, or compromise brain function in Alzheimer’s and other neuropathological diseases.

## 5. Conclusions

The major outcome of this study was the discovery that the pattern of *AQP* expression in the brain is more diverse than previously reported, with possible relevance to processes of healthy aging and AD. The further exploration of the expression and function of aquaglyceroporins, and in particular, peroxiporins, in neurological diseases is an area of ongoing research interest. Identifying the sub-cellular localization of AQP channels could provide critical insights into their potential roles in AD pathophysiology. AQPs are of interest as novel therapeutic targets for the treatment of not just AD, but potentially other neurodegenerative diseases that might similarly rely on changes in the expression profiles of peroxiporins as components of homeostatic responses to age and disease-related stressors.

## Figures and Tables

**Figure 1 biomedicines-11-00770-f001:**
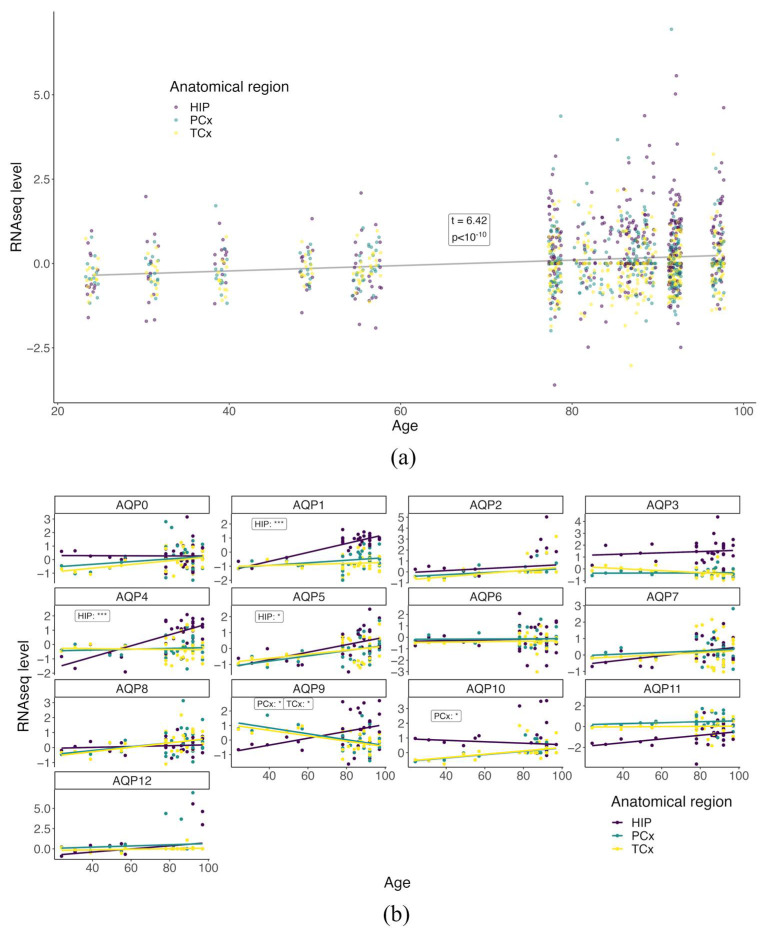
AQP expression profiles show region-specific changes with age in healthy human brains. (**a**) Mixed-effect linear plot testing the relationship between all AQP RNAseq levels and age in the HIP (purple), PCx (green) and TCx (yellow) brain regions. Each point represents the RNAseq level determined for a given gene in a single subject for the selected brain region: HIP (purple), PCx (green) and TCx (yellow). *t*-values represent the test statistic associated with the comparison of the two means using two-sample *t*-tests, with positive values indicating a larger average RNAseq. (**b**) Linear regression plots of individual AQP channels and RNAseq levels as a function of age separated by anatomical areas with a significance change in trends are shown, respectively, as *** for *p* < 0.001 and * for *p* < 0.05. The *p* values for regressions that had significant non-zero slopes are listed as inset text boxes within each figure panel.

**Figure 2 biomedicines-11-00770-f002:**
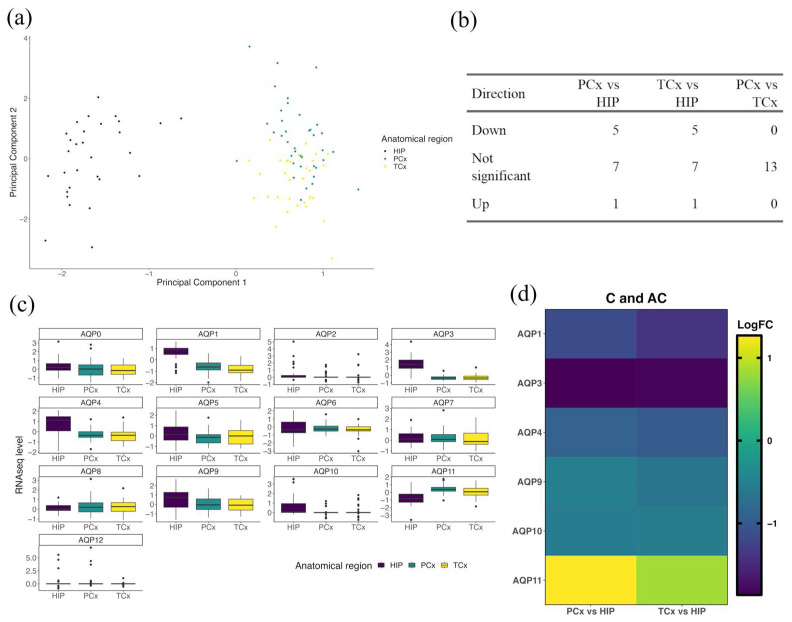
AQP gene expression profiles in normal aging show differences between HIP and cortical regions. (**a**) Supervised cluster analysis of AQP probes in the HIP (purple), PCx (green) and TCx (yellow). Principle components are functions of the probes loaded (Appendix A). (**b**) Summary table of gene expression directions of change and (**c**) corresponding plots of RNAseq levels of all AQP probes in HIP, PCx and TCx. (**d**) Heat map representing differential expression analyses of AQP genes in the PCx and TCx vs. the HIP for healthy patients at all ages (C and AC). LogFC represents the log fold change in gene expression. Only genes with significant changes in LogFC are shown (details in Table 2).

**Figure 3 biomedicines-11-00770-f003:**
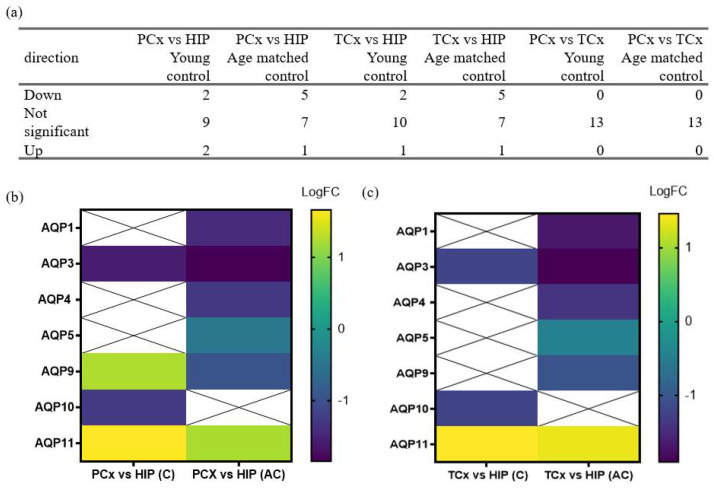
AQP expression profiles differ between non-AD age groups in cortical regions in comparison to the hippocampal formation. (**a**) Summary table of the frequency of occurrence of directions of change in gene expression for non-AD groups (C and AC) in the PCx, TCx and HIP. (**b**,**c**) Heat maps present the results of differential expression analyses of AQP genes in the (**b**) PCx and (**c**) TCx as compared to HIP in non-AD patients, grouped by age. LogFC represents the log fold change in gene expression in the PCx and TCx compared to the HIP, as detailed in Table 3 and Table 4. Squares marked with ‘X’ indicate no significant LogFC change in the respective genes.

**Figure 4 biomedicines-11-00770-f004:**
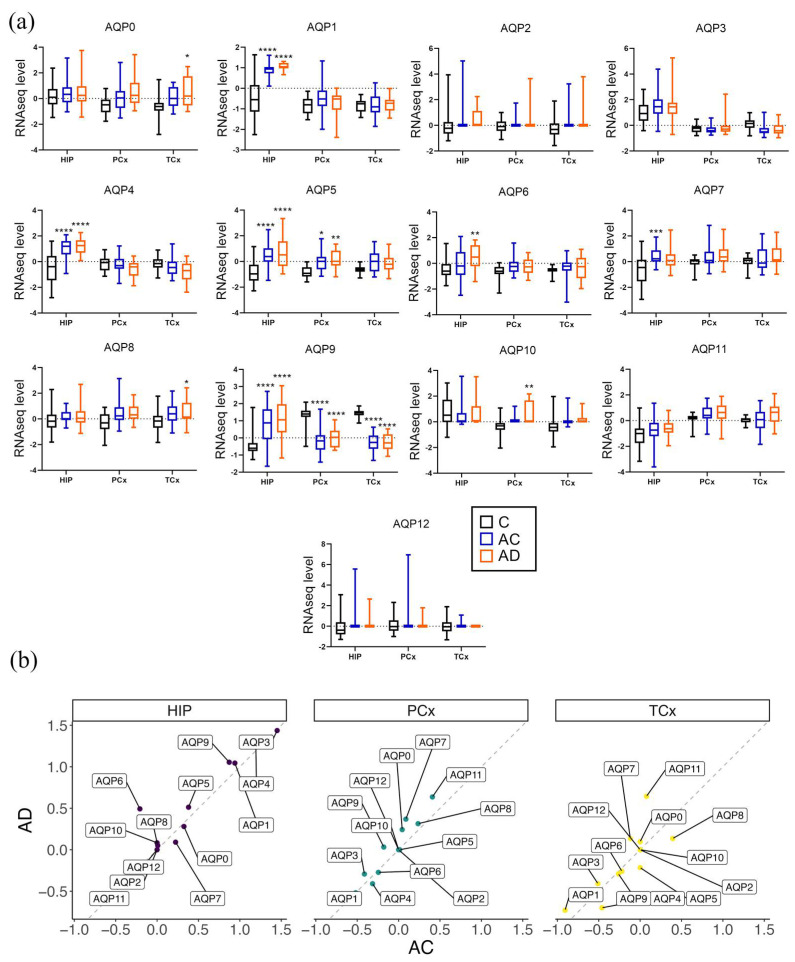
Associations between AD status and AQP transcript levels in the human brain. (**a**) Results of multiple comparison analyses of AQP RNAseq levels in the three groups, young control group (C), age control group (AC) and the AD group, are represented as box and whisker plots showing median, min and max values. Statistical comparisons for AD were carried out with reference to the (C) and (AC) groups, showing **** for *p* < 0.0001, *** for *p* < 0.001, ** for *p* < 0.01 and * for *p* < 0.05. (**b**) The average trend comparison plot of AQP expression profiles in the AC vs. AD groups. Each point represents the average logFC mean expression for each gene and region, for AC vs. AD. The dotted lines represent identity (y = x) in the average transcript levels between the AC and AD groups; points above the line are higher in AD and points below are higher in the AC groups.

**Table 1 biomedicines-11-00770-t001:** Demographic data for human subjects. Age, education and the Braak staging level are shown as median values; numbers in parentheses (1st, 3rd) indicate the 1st and 3rd quartile values. Statistical significant differences between age ranges were evaluated via a non-parametric Mann–Whitney U test; **** indicates *p* < 0.0001 as compared to the young control group; * indicates *p* < 0.05 for AD versus the aged control group; n is the number of subjects.

Groups	n	Age (yrs)	Education	Braak Stage
Young control (C)	6	Range: 24–57 Median: 44 (31, 55)	N/A	N/A
Aged control (AC)	29	Range: 78–99 **** Median: 86 (78.5, 89)	Median: 15 (12, 16)	Median: 3 (2, 3.5)
Probable Alzheimer’s disease (AD)	11	Range: 79–99+ **** Median: 87 (85–91.5)	Median: 14 (12, 16)	Median: 5 (2, 6) *

**Table 2 biomedicines-11-00770-t002:** AQP transcript expression levels in cortical regions differ from those in the hippocampal formation. Differential expression analysis of AQPs in the (i) PCx and (ii) TCx as compared with HIP for subjects in the combined C and AC groups. The average expression (AveExpr) was determined by averaging all RNAseq levels over all probes used for the indicated gene. t-values represent the test statistic associated with the comparison of the two means using two-sample *t*-tests, with positive values indicating a larger average RNAseq in the cortical region compared to the hippocampus. *p* values were adjusted (Adj P) for multiple tests using the Bonferroni correction, with significant differences indicated as *p* < 0.05.

(i) PCx	Gene	logFC	AveExpr	t	*p* Value	adj. *p* Val
Higher expression						
	AQP11	1.2660	−0.1312	6.8948	<0.001	<0.001
Lower expression						
	AQP3	−1.8606	0.2854	−13.497	<0.001	<0.001
	AQP1	−1.1530	−0.2480	−8.046	<0.001	<0.001
	AQP4	−0.8933	0.0305	−4.396	<0.001	1 × 10^−4^
	AQP10	−0.5890	0.2891	−3.164	0.0022	0.0058
	AQP9	−0.5273	0.1916	−2.441	0.017	0.0368
No significant difference						
	AQP5	−0.3550	−0.0021	−1.954	0.0544	0.0911
	AQP2	−0.3983	0.2462	−1.940	0.0561	0.0911
	AQP0	−0.2684	0.0776	−1.362	0.1773	0.2561
	AQP8	0.1785	0.2422	1.0513	0.2964	0.3854
	AQP12	0.1078	0.2901	0.3748	0.7088	0.8377
	AQP7	0.0299	0.2040	0.1616	0.8721	0.9448
	AQP6	0.0129	−0.2175	0.0623	0.9505	0.9505
**(ii) TCx**	**Gene**	**logFC**	**AveExpr**	**t**	***p* Value**	**adj. *p* Val**
Higher expression						
	AQP11	0.8442	-0.1312	4.6514	<0.001	<0.001
Lower expression						
	AQP3	−1.8251	0.2854	−13.39	<0.001	<0.001
	AQP1	−1.3989	−0.2480	−9.876	<0.001	<0.001
	AQP4	−0.9932	0.0305	−4.945	<0.001	<0.001
	AQP9	−0.6515	0.1916	−3.051	0.0031	0.0082
	AQP10	−0.5451	0.2891	−2.963	0.0041	0.0088
No significant difference						
	AQP0	−0.4008	0.0776	−2.057	0.0431	0.0801
	AQP5	−0.3258	−0.0021	−1.814	0.0737	0.1197
	AQP2	−0.3557	0.2462	−1.753	0.0837	0.1208
	AQP12	−0.3756	0.2901	−1.321	0.1904	0.2475
	AQP8	0.1857	0.2422	1.106	0.2721	0.3216
	AQP6	−0.2120	−0.2175	−1.039	0.302	0.3271
	AQP7	−0.0685	0.2040	−0.375	0.7086	0.7086

**Table 3 biomedicines-11-00770-t003:** Ratios of quantified *AQP* transcript levels in parietal cortex as compared to the hippocampal formation differ with age. The average expression (AveExpr) levels were determined by averaging the RNAseq levels for all probes used for a given gene. *t*-values represent the test statistic generated by a comparison of the means using two-sample *t*-tests, with positive values indicating a larger average value for the RNAseq level in the parietal cortex region as compared to the hippocampus. P values were adjusted (Adj P) for multiple tests using the Bonferroni correction, with significant differences indicated by *p* < 0.05.

(i) C	Gene	logFC	AveExpr	t	*p* Value	adj. *p* Val
Higher expression						
	AQP11	1.6950	−0.1312	3.9866	1 × 10^−4^	8 × 10^−4^
	AQP9	1.2396	0.1916	2.7069	0.0079	0.0257
Lower expression						
	AQP3	−1.5951	0.2854	−4.4059	<0.001	3 × 10^−3^
	AQP10	−1.2611	0.2891	−3.1242	0.0023	0.01
No significant difference						
	AQP0	−0.7967	0.0776	−1.6654	0.0988	0.2568
	AQP4	0.6127	0.0305	1.5691	0.1196	0.2591
	AQP7	0.4365	0.2040	1.0265	0.307	0.5701
	AQP2	−0.3546	0.2462	−0.7644	0.4463	0.7253
	AQP6	0.2345	−0.2175	0.5093	0.6116	0.8834
	AQP12	0.1776	0.2901	0.2705	0.7873	0.9189
	AQP8	−0.0834	0.2422	−0.2207	0.8258	0.9189
	AQP5	0.0477	−0.0021	0.1169	0.9072	0.9189
	AQP1	−0.0299	−0.2480	−0.1020	0.9189	0.9189
**(ii) AC**	**Gene**	**logFC**	**AveExpr**	**t**	***p* Value**	**adj. *p* Val**
Higher expression						
	AQP11	1.2161	−0.1312	6.0634	<0.001	<0.001
Lower expression						
	AQP3	−1.8470	0.2854	−10.815	<0.001	<0.001
	AQP1	−1.4439	−0.2480	−10.439	<0.001	<0.001
	AQP4	−1.2894	0.0305	−7.0001	<0.001	<0.001
	AQP9	−0.9655	0.1916	−4.4691	<0.001	1 × 10^−4^
	AQP5	−0.4671	−0.0021	−2.4253	0.017	0.0368
No significant difference						
	AQP10	−0.3984	0.2891	−2.0920	0.0388	0.0721
	AQP2	−0.3731	0.2462	−1.7051	0.0911	0.148
	AQP8	0.2499	0.2422	1.4010	0.1641	0.2371
	AQP0	−0.0976	0.0776	−0.4325	0.6663	0.8245
	AQP12	0.1206	0.2901	0.3895	0.6977	0.8245
	AQP7	−0.0571	0.2040	−0.2847	0.7764	0.8411
	AQP6	−0.0128	−0.2175	−0.0587	0.9533	0.9533

**Table 4 biomedicines-11-00770-t004:** Ratios of quantified AQP transcript levels in the temporal cortex as compared to the hippocampal formation differ with age. The average expression (AveExpr) levels were determined by averaging the RNAseq levels for all probes used for a given gene. *t*-values represent the test statistic associated with the comparison of means using two-sample *t*-tests, with positive values indicating a larger average RNAseq in the cortical region as compared to the hippocampus. P values were adjusted (Adj P) for multiple tests using the Bonferroni correction, with significant differences indicated by values *p* < 0.05.

(i) C	Gene	logFC	AveExpr	t	*p* Value	adj. *p* Val
Higher expression						
	AQP11	1.4622	−0.1312	3.4391	8 × 10^−4^	0.0074
Lower expression						
	AQP3	−1.2113	0.2854	−3.3457	0.0011	0.0074
	AQP10	−1.2381	0.2891	−3.0671	0.0027	0.0119
No significant difference						
	AQP9	1.0602	0.1916	2.3150	0.0225	0.0732
	AQP4	0.7552	0.0305	1.9340	0.0558	0.1209
	AQP0	−0.9250	0.0776	−1.9335	0.0558	0.1209
	AQP2	−0.5220	0.2462	−1.1253	0.263	0.4884
	AQP7	0.2543	0.2040	0.5980	0.5511	0.8955
	AQP8	−0.1370	0.2422	−0.3624	0.7178	0.9279
	AQP1	−0.0974	−0.2480	−0.3323	0.7403	0.9279
	AQP5	0.1005	−0.0021	0.2461	0.8061	0.9279
	AQP12	−0.0745	0.2901	−0.1134	0.9099	0.9279
	AQP6	−0.0418	−0.2175	−0.0908	0.9279	0.9279
**(ii) AC**	**Gene**	**logFC**	**AveExpr**	**t**	***p* Value**	**adj. *p* Val**
Higher expression						
	AQP11	0.7296	−0.1312	3.6732	4 × 10^−4^	0.001
Lower expression						
	AQP1	−1.7035	−0.2480	−12.435	<0.001	<0.001
	AQP3	−1.9022	0.2854	−11.247	<0.001	<0.001
	AQP4	−1.4108	0.0305	−7.7335	<0.001	<0.001
	AQP9	−1.0480	0.1916	−4.8983	<0.001	<0.001
	AQP5	−0.4470	−0.0021	−2.3440	0.0209	0.0454
No significant difference						
	AQP10	−0.3690	0.2891	−1.9567	0.053	0.0984
	AQP8	0.2502	0.2422	1.4164	0.1596	0.2227
	AQP12	−0.4251	0.2901	−1.3858	0.1687	0.2227
	AQP2	−0.2985	0.2462	−1.3772	0.1713	0.2227
	AQP0	−0.2390	0.0776	−1.0695	0.2873	0.3395
	AQP6	−0.2083	−0.2175	−0.9685	0.335	0.3629
	AQP7	−0.1479	0.2040	−0.7444	0.4583	0.4583

## Data Availability

Data are available online on the Allen Brain Atlas. Link: https://portal.brain-map.org/, accessed on 20 December 2022.

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
