# Peer review of "Unexpected Classes of Aquaporin Channels Detected by Transcriptomic Analysis in Human Brain Are Associated with Both Patient Age and Alzheimer’s Disease Status"

_biomedicines, 2023, doi:10.3390/biomedicines11030770_

Round 1

Reviewer 1 Report

In the current paper, Amro et al. investigated data from the Human brain atlas database and the TBI database analyzing the transcript expression  of all  human aquaporins in healthy and Alzheimer’s disease brains. In particular the study focused on hippocampus, parietal cortex and temporal cortex, three brain regions affected by Alzheimer’s disease. Firstly, they discovered a different AQP expression profiles in young and old healthy brain regions and secondly, they find out a probable and new pattern of AQP expression in Alzheimer’s disease brains. Overall, the paper is well written although the limitation about the low n values available for human samples and also the future perspective are interesting. In my opinion the paper could be published after making a minor revision. I just have some comments and a suggestion.

- Pay attention to the use of abbreviations because there are many abbreviation not explained in the text and so the reading became difficult. I suggest to use the abbreviation for the first time between parenthesis and then only the abbreviation. Line 39: Aβ; Line 50: H2O2 ; Line 61: CSF; Line 68: CNS

- Line 39: “ Importantly, Aβ and tau-based evaluations indicate that the disease spreads through neighboring anatomical areas [9,10] by a process proposed to involve networks of astrocytes and microglia [11].” Could you add a sentence to list the brain regions selected for relevance in AD pathology?

- Line 52: Also AQP9 is a peroxiporins (S. Watanabe, C.S. Moniaga, S. Nielsen, M. Hara-Chikuma, Aquaporin-9 facilitates membrane transport of hydrogen peroxide in mammalian cells, Biochem.Biophys. Res. Commun. 471 (2016) 191e197). Please, add AQP9 with a reference.

- Line 101: “ In brief, data were normalised within each brain dissection batch via multivariate local regression in accord with detailed method documents.” Could you list in this line the brain dissection batches considered?

- Line 107: “ Data were collected from four brain regions (Supplementary table 1); however, given the anatomical areas of interest affected in AD as outlined by the Braak staging for disease spread[9], data associated with white matter was excluded from analysis in the current study.” Writing about the brain regions considered only in this paragraph of the Materials and Methods section is confusing. Reading this sentence I could misunderstand that you considered that brain regions only for the young control group.

- Remember to write Supplementary Table S1 (line 107), S2 (line 130) and S3 (line 176) because Table 1, and so on, are already present.

- Please, add a legend to Table S2.

- Please, complete the legend of Table S3 adding abbreviations for F, M, N. Why there is a column for Ethnicity only in table S3a and a column for Education only in table S3c? Are these parameters essential for the study?

- Please, adjust the resolution of  all the figures

- The legend of Fig.1 is not fully completed, please add statistic in a). In the graph of AQP10 there is a box. Is also the expression of AQP10 statistically significant? I cannot find this point in the results.

- Fig. 2: in figure 2, PC and TC are separated and colored differently. Have you considered PC and TC as a single group during the analysis? If yes, I suggest to underline it in the legend.

- Line 383: “Of these channels, peroxiporins AQPs 0, 1 and 11 (which are permeable to H2O2)…”Please, add references for H2O2 permeability of AQPs 0, 1 and 11.

- Line 461: “Limitations of this work are the modest n values available for human brain RNAseq data, which likely contributed to the lack of significant differences in expression profiles between the AD and the AC cohorts…” I just have a curiosity and also an advice: have you ever tried to analyze separately female and male patients? Or have you tried to analyze AD and AC cohorts with the same female/male ratio? Because Alzheimer’s disease is a predominant female pathology (Pike CJ. Sex and the development of Alzheimer's disease. J Neurosci Res. 2017 Jan 2;95(1-2):671-680. doi: 10.1002/jnr.23827. PMID: 27870425; PMCID: PMC5120614) and the female/male ratio in AD and AC cohorts is really different (AD: 3 female patients on 11, AC: 14 female patiens on 29).

Author Response

We appreciate the constructive comments made by all reviewers and have incorporated the requested changes in the revised MS. The edits are summarized below and highlighted in blue font in the pdf version of the manuscript, and we believe have strengthened the paper. We thank the reviewers for their careful evaluations and useful feedback.  

Pay attention to the use of abbreviations because there are many abbreviation not explained in the text and so the reading became difficult. I suggest to use the abbreviation for the first time between parenthesis and then only the abbreviation. Line 39: Aβ; Line 50: H2O2 ; Line 61: CSF; Line 68: CNS

     These helpful details have been addressed as suggested; the edited text is highlighted in blue font in the pdf version of the revised MS. 

Line 39: “ Importantly, Aβ and tau-based evaluations indicate that the disease spreads through neighboring anatomical areas [9,10] by a process proposed to involve networks of astrocytes and microglia [11].” Could you add a sentence to list the brain regions selected for relevance in AD pathology?

     The requested information has been added as new text to the Introduction (lines 39-44), which reads: “Importantly, amyloid beta (Aβ) and tau-based evaluations indicate that the disease spreads through neighbouring anatomical areas beginning at the hippocampal formation and areas of the temporal (e.g. entorhinal cortex) and parietal (e.g.. retrosplenial cortex; posterior parietal cortex; precuneus) lobes in preclinical stages of the disease before spreading to additional regions (e.g. prefrontal cortex; amygdala) as individuals become symptomatic (9, 10) in a process proposed to involve networks of astrocytes and microglia (11).”

- Line 52: Also AQP9 is a peroxiporin (S. Watanabe, C.S. Moniaga, S. Nielsen, M. Hara-Chikuma, Aquaporin-9 facilitates membrane transport of hydrogen peroxide in mammalian cells, Biochem.Biophys. Res. Commun. 471 (2016) 191e197). Please, add AQP9 with a reference. 

     The added text and reference in the Introduction (lines 58-59) now reads: ‘...AQP9 also has been shown to permeate H2O2 in mice (26).’

- Line 101: “ In brief, data were normalised within each brain dissection batch via multivariate local regression in accord with detailed method documents.” Could you list in this line the brain dissection batches considered?

     The edited Materials and Methods (lines 109-110) now reads: ‘In brief, brains were processed serially with multiple sample batches submitted per brain analysed. Data were then normalised to an internal control within each brain dissection batch via multivariate local regression, in accord with detailed method documents (https://help.brain-map.org/display/humanbrain/Documentation).’

- Line 107: “Data were collected from four brain regions ...” Writing about the brain regions considered only in this paragraph of the Materials and Methods section is confusing. Reading this sentence I could misunderstand that you considered that brain regions only for the young control group.

     We agreed this point needed to be clarified, and revised the text to read (lines 116-122): ‘Data from this database, in addition to the Institute of Aging, Dementia and TBI database, were collected from four brain regions (Supplementary table 1) for all age groups; however, given the anatomical areas of interest affected in AD as outlined by the Braak staging for disease spread (9), data associated with white matter was excluded from analysis in the current study (11).

- Remember to write Supplementary Table S1 (line 107), S2 (line 130) and S3 (line 176) because Table 1, and so on, are already present.

     Done as requested

- Please, add a legend to Table S2.

     Done as requested; this required only the addition of an informative title.

- Please, complete the legend of Table S3 adding abbreviations for F, M, N. Why there is a column for Ethnicity only in table S3a and a column for Education only in table S3c? Are these parameters essential for the study?

     Done as requested. The Ethnicity and Education status data are not essential for the study as presented here, but might be relevant to other researchers using our results as a starting point for their own work.

- Please, adjust the resolution of all the figures

     All figures have been updated to the maximum possible resolution.

- The legend of Fig.1 is not fully completed, please add statistic in a). In the graph of AQP10 there is a box. Is also the expression of AQP10 statistically significant? I cannot find this point in the results.

     Done as requested; the statistical analysis is stated (lines 244-246). Text has been added (lines 209-210) to clarify: ‘AQP5 and 10 gene expression profiles also increased with age in the HIP and PCx, respectively (P<0.05), novel AQP channels not previously identified in the human brain.’

- Fig. 2: in figure 2, PC and TC are separated and colored differently. Have you considered PC and TC as a single group during the analysis? If yes, I suggest to underline it in the legend.

     We suggest that using separate colors for PCx and TCx allows data to be compiled into a single figure which simultaneously shows both the similarities and differences between these regions, and at the same time maintains visual consistency with the other figures. Though more similar to each other than to hippocampus, these cortical areas do not behave identically, and have been shown to be differentially affected in AD and in healthy aging (Bakkor et al. 2013, DOI: 10.1016/j.neuroimage.2013.02.059), suggesting that it is appropriate to keep them as distinct groups.

- Line 383: “Of these channels, peroxiporins AQPs 0, 1 and 11 (which are permeable to H2O2)…”Please, add references for H2O2 permeability of AQPs 0, 1 and 11.

     In the original MS, this line referred to peroxiporins AQP0, 6 and 8 (not 1 and 11); nonetheless in accord with the spirit of the request, we have now provided 3 references for the statement (line 419) as recommended.

-  I just have a curiosity and also an advice: have you ever tried to analyze separately female and male patients? Or have you tried to analyze AD and AC cohorts with the same female/male ratio? Because Alzheimer’s disease is a predominant female pathology (Pike CJ. Sex and the development of Alzheimer's disease. J Neurosci Res. 2017 Jan 2;95(1-2):671-680. doi: 10.1002/jnr.23827. PMID: 27870425; PMCID: PMC5120614) and the female/male ratio in AD and AC cohorts is really different (AD: 3 female patients on 11, AC: 14 female patiens on 29).

     We appreciate the Reviewer’s interesting comment on the gender implications for AD progression. Tackling this idea quantitatively is unfortunately not feasible at this time using the current database, due to the small sample size for females in the AD group (n = 3). However, this fascinating suggestion could be one of the reasons why no significant differences were observed between the AD and AC groups. We have added this consideration to the Discussion section (lines 500-511), which reads: ‘Another important limitation that might have influenced the absence of statistically significant differences between the AD and AC groups was the low representation of female donors in the AD group (n=3) as compared to AC (n=11). The risk for development and progression of AD in females on average is higher than males but depends on estrogen levels (74); a re-analysis of the AD and AC groups segregated by gender and hormone therapy status might reveal important correlations with AQP expression profiles that merit exploration when expanded database information becomes available. An additional limitation is the need to confirm AQP expression at the protein level, to determine whether the expression profile changes determined by transcript analyses are reflected by changes at the protein level. Probing the functionality of the proteins then will be an essential next step towards identifying possible novel targets for therapeutic intervention in AD.’

Reviewer 2 Report

Altered expression of known brain Aquaporin (AQP) channels 1, 4 and 9 has been corre

lated with neuropathological AD progression, but possible roles of other AQP classes in neurologi

cal disease remains understudied. Levels of transcripts of all thirteen human AQP subtypes were

compared in healthy and Alzheimer’s Disease (AD) brains by statistical analyses of microarray

RNAseq expression data from the Allen Brain Atlas database. Previously unreported, AQPs 0, 6, 14

and 10, are present in human brain at the transcript level. Three AD-affected brain regions, hippo

campus (HIP), parietal cortex (PCx) and temporal cortex (TCx), were assessed in three subgroups:

young controls (n=6, aged 24-57); aged controls (n=26, aged 78-99); and an AD cohort (n=12, aged

79-99). Significant positive correlation was seen for AQP transcript levels as a function of

subject age in years. Differential expression correlated with brain region, age, and AD diagnosis,

particularly between the HIP and cortical regions. Of these, AQPs 0 and 8 were 

increased in TCx, and AQP6 in HIP suggesting a role of AQPs in AD-related oxidative stress. These results suggest the expression profile of AQP channels in the human brain is more diverse than previously thought, and transcript levels are influenced by both age and AD status. Associations between reactive oxygen stress and neurodegenerative disease risk highlight AQPs 0, 6, 8 and 10 as potential therapeutic targets.

Author Response

     The Reviewer has presented a crisp concise summary of the main findings without noting any concerns or corrections; we appreciate the confirmation that the outcomes presented were found  to be clear and convincing.

Reviewer 3 Report

Amro et al. examined in their paper expression levels of human aquaporin genes in human brain using RNAseq expression data from Allen Brain Atlas and data of the ATC study. The analysis solely depends on the expression data extracted from the data base. Thus the manuscript does not provide new data, but a new analysis of alreading existing data. In the discussion (first paragraph), the authors argue that they have demonstrated presence of additional AQP expression in human brain and identified novel RNAseq signals for AQP in human brain. But these data have been extracted from data bases and therefore these AQP expressions have been found before. Important limitations (as also pointed out by the authors) are the low n number (especially in the young control group). The most interesting comparisons would be between the groups AC and AD as well as young and aged controls (C vs AC), in order to a show a possible correlation between AQP expression changes and AD. C vs AC is shown in Figure 4 panel a); The interesting comparison in order to find indication for possible AD associated changes in gene expression would be comparison between AC and AD groups: but this is not done in Figure 4 (?). From the data shown in panel a) of Figure 4 it appears that there are not many (if at all) statistically significant differences between AC and AD. The paper would much more interesting if new date would be presented beside the analysis of already existing datasets. E.g. comfirmation of the expression data by RT-PCR analysis of independnet samples; or Western blot analysis to examine whether changes observed at the transcriptional levels also leads to changes at the protein level. line 195/196: AQP5 is mentioned as a water channel "not previously identified in the human bran": however the data present in the RNAseq data set: thus it has been identified in human brain before (see comment above). Figure 1: low resolution, p values in panel B are not readable (it also seems that not p values are shown in the boxes in panel B, but the brain region and asterisks) line 423-326: Here it is stated that AQP3 and AQP11 levels in AD were compared to AC and C; however in the Figure legend to Figure 4, it is stated that data were only compared to C.

Author Response

  1. The analysis solely depends on the expression data extracted from the data base. Thus the manuscript does not provide new data, but a new analysis of alreading existing data....The authors argue that they have demonstrated presence of additional AQP expression in human brain and identified novel RNAseq signals for AQP in human brain. But these data have been extracted from data bases and therefore these AQP expressions have been found before.

     We thank the Reviewer for providing us with an important perspective on the need to ensure that the experimental strategy is clearly explained for a broad readership. The Reviewer's main concerns on novelty of the work stemmed from the fact that our study was done by harvesting transcriptomic data from a public domain database for in-depth analysis. The presence of a mass-throughput generated database does not mean that discoveries have already been made; in fact, the vast majority of the raw information contained in the database remains to be assessed. The expression profiles we report here have not been previously assessed;  new AQPs in the human brain described here have not previously been described, and our paper is first to explore this gap, presenting a novel discovery and contribution to the field that we anticipate will open a new area of research. To help clarify this important concept for our readers, we added a reference to high profile paper that used a comparable strategy to study another neuropathology, autism (new reference 48 ; "Transcriptomic Analysis of Autistic Brain Reveals Convergent Molecular Pathology", doi: 10.1038/nature10110).  We also added the following text to the paper (lines 100-102): ‘Notably, the Allen Brain Atlas serves as a substantial archive of collated RNAseq data that remains to be analyzed; this public domain database is invaluable for enabling novel discoveries, as demonstrated in previous work (49).’

  1. Comparison between the AC and AD group (“The most interesting comparisons would be between the groups AC and AD as well as young and aged controls in order to a show a possible correlation between AQP expression changes and AD. The interesting comparison between AC and AD groups is not done in Figure 4”)

     We apologize for the lack of clarity in the original Fig 4 legend. In Figure 4, all data for AQP genes in the AD group were compared against the control groups (C and AC). We have amended the figure legend (lines 377 -378). No significant differences between AD and AC groups for any AQP gene were detected though clear trends were described in the original MS text (lines 352-358),  and possible reasons for the lack of clear differences have been highlighted in the Discussion (lines 495-510).

  1. “The paper would much more interesting if new date would be presented beside the analysis of already existing datasets. E.g. comfirmation of the expression data by RT-PCR analysis of independnet samples; or Western blot analysis to examine whether changes observed at the transcriptional levels also leads to changes at the protein level.”

     We understand the Reviewer was concerned that the MS appeared to lack novelty due to use of a database, which we have addressed in response to point 1 above. Limitations of the current work have been expanded in the Discussion (lines 499-510) to include a need to confirm protein expression (as well as localization and functionality as the next steps in progressing this field; the text now reads: “Limitations of this work are the modest n values available for human brain RNAseq data, which likely contributed to the lack of significant differences in expression profiles between the AD and the AC cohorts (though trends towards increased RNAseq levels were apparent in the disease group), and that age-dependent effects on AQP expression are likely to overlap with the disease pathology. Another important limitation that might have influenced the absence of statistically significant differences between the AD and AC groups was the low representation of female donors in the AD group (n=3) as compared to AC (n=11). The risk for development and progression of AD in females on average is higher than males but depends on estrogen levels (74); a re-analysis of the AD and AC groups segregated by gender and hormone therapy status might reveal important correlations with AQP expression profiles that merit exploration when expanded database information becomes available. An additional limitation is the need to confirm AQP expression at the protein level, to determine whether the expression profile changes determined by transcript analyses are reflected by changes at the protein level. Probing the functionality of the proteins then will be an essential next step towards identifying possible novel targets for therapeutic intervention in AD.”

Minor comment: AQP5 is mentioned as a water channel "not previously identified in the human bran": however the data present in the RNAseq data set: thus it has been identified in human brain before.

     As noted above, the publicly available database is a pooled resource of RNAseq data from mass screening with no analysis or interpretations made. AQP5 and others noted in this MS have not been reported in the human brain.

Figure 1: low resolution, p values in panel B are not readable.

     We have attempted to maximize the resolution of the Figures,  also requested by Reviewer 1.  P values should now be readable.

line 423-326: Here it is stated that AQP3 and AQP11 levels in AD were compared to AC and C; however in the Figure legend to Figure 4, it is stated that data were only compared to C.

    Data for AD in Fig 4a were compared to both C and AC groups. We apologize for not mentioning AC in the original figure legend but have now done so, and thank the Reviewer for catching this error.

Round 2

Reviewer 3 Report

The authors have addressed the questions raised by this reviewer to a large extent.